# Behavioral Modifications in Children after Repeated Sedation with Nitrous Oxide for Dental Treatment: A Retrospective Study

**DOI:** 10.3390/ijerph20054037

**Published:** 2023-02-24

**Authors:** Annelyse Garret-Bernardin, Paola Festa, Giorgio Matarazzo, Arina Vinereanu, Francesco Aristei, Tina Gentile, Simone Piga, Elena Bendinelli, Maria Grazia Cagetti, Angela Galeotti

**Affiliations:** 1Dentistry Unit, Bambino Gesù Children’s Hospital, IRCCS, Viale Ferdinando Baldelli 41, 00146 Rome, Italy; 2Department of Pedodontics, Faculty of Dental Medicine, ‘Carol Davila’ University of Medicine and Pharmacy, 032799 Bucharest, Romania; 3Unit of Clinical Epidemiology, Bambino Gesù Children’s Hospital, IRCCS, Piazza Sant’Onofrio 4, 00165 Rome, Italy; 4Department of Surgery and Translational Medicine, University of Florence, 50121 Firenze, Italy; 5Department of Biomedical, Surgical and Dental Sciences, University of Milan, Via Beldiletto 1, 20142 Milan, Italy

**Keywords:** nitrous oxide, conscious sedation, behavior, dental anxiety, pediatric dentistry

## Abstract

Sedation with nitrous oxide (N_2_O) has been widely used as a viable alternative to general anesthesia to perform dental treatments in uncooperative or anxious children. The purpose of this retrospective study is to assess if repeated sedations with N_2_O can improve collaboration of uncooperative children. The medical records of 650 children, aged between 3 and 14 years, who underwent at least two sedations, were consulted. Differences in the Venham score during the first sedation and subsequent sedations were collected. After removal incomplete records, 577 children’s records (309 males and 268 females) were analyzed. The Venham score decreased both during each sedation and with repeated sedations (*p* < 0.01 for both comparisons). In particular, a significant reduction of the Venham score was observed at the first contact with the dentist, with a mean score ranging from 1.56 ± 1.46 to 1.16 ± 1.37, comparing the first and the second sedation, and from 1.65 ± 1.43 to 1.06 ± 1.30, comparing the first with the third sedation (*p* < 0.01). The reduction in the Venham score was recorded in both healthy and physically impaired patients, and it was significantly greater in older children than in younger children (*p* < 0.01). In conclusion, uncooperative children with or without physical impairments can be successfully treated with N_2_O sedation in order to increase their confidence in dental procedures.

## 1. Introduction

Conscious sedation with nitrous oxide (N_2_O) is widely used to improve patient cooperation during treatment in various medical/surgical fields due to its ability to induce both analgesia and anxiety control [1]. Inhalation of N_2_O and oxygen (O_2_) has been extensively studied in dentistry to reduce the anxiety that may be associated with dental treatments, and its short-term effects have been well documented for different concentrations of the two gases [2].

In children, successful dental treatments can be compromised by poor compliance [3]. Behavioral management techniques, such as desensitization, positive-negative reinforcement, tell-show-do, are the first attempt to overcome unwanted behaviors during dental procedures. However, this approach can be sufficiently effective for some children, but not for all [4]. Children who are extremely fearful or have intellectual deficits often require advanced behavioral guidance techniques, such conscious sedation or general anesthesia [5,6]. In the past decades, general anesthesia was the most used technique to overcome behavioral barriers to treatment, even though it can cause negative health outcomes and is an expensive procedure [7,8].

According to the Italian guidelines on conscious sedation in dentistry, published in May 2021, N_2_O sedation is recommended as a standard procedure in pediatric patients to reduce the use of general anesthesia [9,10].

In recent years, sedation with N_2_O has been successfully performed in pediatric dentistry to manage opposition, the gag reflex and anxiety, and to improve cooperation [11,12]. Moreover, children treated with nitrous oxide sedation have been shown to have lower postoperative anxiety levels in comparison to those treated under general anesthesia [11,13]. In fact, N_2_O sedation may be repeatedly used in order to decrease anxiety and mitigate the impact of dental treatments and the consequent difficult behavior [12]. Despite much evidence on the effectiveness of this inhalation sedation technique, long-term effects on patient behavior after receiving more than one sedation have rarely been described [1].

The purpose of this retrospective study is to evaluate, through examination of medical records, whether repeated conscious sedations with N_2_O for dental treatments improve the behavior of uncooperative children.

## 2. Materials and Methods

A retrospective study was designed in accordance with the Helsinki declaration and registered at ClincalTrials.gov (accessed on 8 February 2023). It was conducted at the Dentistry Unit of Bambino Gesù Children’s Hospital, IRCCS, in Rome (Italy). Approval was achieved from the local ethical committee (2176_OPBG_2020). This study was realized within OSCAR project: ERASMUS+ Project 2019- 1-RO01-KA202-063820.

### 2.1. Subjects

The medical records of patients who underwent conscious sedation with N_2_O at the Dental Unit of Bambino Gesù Children’s Hospital, IRCCS, in Rome (Italy) between January 2013 and December 2019 were considered.

Inclusion criteria were: records on children aged between 3 and 14 years, uncooperative during the first attempt at dental treatment without sedation, undergoing at least two sedations for invasive dental treatments (restoration/endodontic/extraction) performed by the same operator.

The exclusion criteria were: records of children who had had previous experience of conscious sedation with N_2_O or other type of sedation or previous dental treatment in another facility, records with incomplete data.

In the sedation session, heart rate, oxygen saturation, and blood pressure are monitored regularly at the beginning of the dental treatment, every 10 min during the procedure, and at the end of the treatment. Dose and time of administration of local anesthesia and inspired concentration of O_2_ and N_2_O are also recorded, as well as the level of cooperation, using the modified Venham scale (Table 1) [14,15,16]. This scale offers a reliable description of the children’s behavior and anxiety expressed by the operator with a score ranging from 0 to 5.

This sedation procedure is standardized for all patients treated at the Dental Unit as described in a previous paper [14].

### 2.2. Sedation Procedure

A mixture of 50% N_2_O/50% O_2_ is administered through a nasal mask, chosen from different shapes and sizes depending on the patient’s age and facial morphology. The gas is delivered by a sedation machine with an internal mixing device (Master Flux Plus Automatic AS3000, Techno-Gaz Industries, Sala Baganza, Italy). The exhaled gas evacuation tube empties externally. An internal ventilation system provides continuous forced recirculation of air in the room after each administration.

### 2.3. Data Collection

The following data were retrospectively and anonymously retrieved from the dental charts and input into a Microsoft Excel 2020 spreadsheet (Microsoft Corporation, Redmond, WA, USA): sex, age, systemic diseases or disabilities, number of sedations, and the patient’s behavior assessed by the dentist using the modified Venham Scale (Table 1) at five steps of each procedure: (1) Tc: first contact with the dentist, (2) T0: at the moment of placing the mask on the nose, (3) T1: at least 3 min after the start of sedation, but before starting any dental treatment, (4) T2: during the administration of local anesthesia, and (5) T3: during the dental treatment.

### 2.4. Statistical Analysis

Detailed statistics and mean values (±Standard Deviation) or median with IQR and ranges were calculated as appropriate for the data distribution. Percentages were reported for categorical or dichotomous variables.

The data analysis was designed to investigate any potential differences in the modified Venham scores registered at each step (Tc, T0, T1, T2, and T3) of each sedation (two or three).

Venham’s scores at Tc (i.e., the first contact with the dentist) were taken as reference for calculating the sample size based on the mean difference, as this phase was considered critical for establishing a trusting and empathic bond with the patient. As no literature data were available, an average reduction in Venham scores between the first and third sedation sessions of at least one mean score point was assumed, based on experience. Based on these assumptions, with an alpha level of 0.05 and a study power of 90%, at least 100 patients with more than two sedations were required.

To assess the normality of the distribution of each variable, a Shapiro–Wilk test was applied. To determine whether different characteristics of the children (age, sex, presence of physical or intellectual impairments) could influence statistical differences in behavior, the chi-square test or Fischer’s exact test for categorical variables and the t-test or ANOVA test or Mann–Whitney test for continuous variables were applied.

The correlation between parameters was assessed with Pearson’s r index or Spearman’s rho when appropriate. Data analysis was carried out with STATA statistical software, version 13.

## 3. Results

A total of 650 medical records of children who underwent at least two sedations were considered. Incomplete records (n = 73) were excluded, leaving 577 records of children (309 males and 268 females) for analysis: 348 patients underwent two sedations and 229 patients underwent three sedations (Figure 1).

The mean age of the children was 6.66 ± 2.23. Of the children, 367 were between 3 and 6 years old, and 210 were between 7 and 14 years old. In term of general health, 95 children were affected by systemic diseases, and of these 51 had an intellectual impairment (Table 2).

The results showed the Venham score decreased progressively both during each sedation and with the repetition of sedations (*p* < 0.001), as shown in Figure 2, Figure 3 and Figure 4.

Figure 2 shows the comparison of Venham scores between the first contact with the dentist and the subsequent steps of the treatment during the first sedation. The mean values of the scores measured at the different steps were significantly lower (*p* < 0.01) than the mean recorded at Tc (mean score 1.56 ± 1.46), reaching the lowest values at T1 (mean score 0.64 ± 1.21) and T2 (mean score 0.61 ± 0.48). However, a slight increase in the score was recorded during the dental treatment (T3 mean score 0.93 ± 1.46).

Figure 3 shows Venham score comparisons between Tc and T2 in the three sedations. A statistically significant decrease in Venham scores was recorded for all comparisons (*p* < 0.01).

Comparisons of Venham scores between the three sessions at Tc and T0 is shown in Figure 4. A statistically significant reduction of the score (*p* < 0.01) was showed from the first contact (Tc) in the first sedation (s1) compared with the following two sedations (s2 and s3) (Figure 4a). Finally, a statistically significant reduction of the score (*p* < 0.01) was also recorded comparing the time of mask placement (T0) in the first sedation (s1) compared to the same step of following sedations (mean score 1.64 and 1.05, respectively, for s2 and s3) (Figure 4b).

No statistically significant differences were found between males and females in the Venham scores (*p* > 0.05) for any times and sedations (data not in table). Consequently, to ensure a more consistent number, the data are not shown by gender.

Although a statistically significant reduction of mean Venham score was recorded in children of the two age groups considered (*p* < 0.01), a greater decrease of almost one point of the Venham scale was recorded in older children (aged 7–14 years) compared to younger (aged 3–6 years). Overall, a reduction in the total average Venham score of 0.9 ± 1.3 was found in children aged 7–14 years and 0.6 ± 1.4 in children aged 3–6 years (*p* = 0.01).

In relation to the health status of children, the Venham score recorded at the first contact with the dentist (Tc) in the first sedation in children with an intellectual impairment and in those with a physical impairment was higher (respectively, 1.8 ± 1.5 and 1.6 ± 1.5) than in those without diseases (1.5 ± 1.5), but the difference was not statistically significant (*p* = 0.57).

However, the mean reduction in the scores recorded at Tc during the first sedation compared to those recorded at the same step of the subsequent sedation (s2) was similar in the three groups (0.4 ± 1.6). Comparing the different sedations, a reduction in the Venham score was found at T3 in all groups, but only for healthy children was the reduction statistically significant (*p* < 0.01) (data not in the table).

## 4. Discussion

Few studies have dealt with behavioral modification in children treated with N_2_O for dental treatment [1,10,17,18]. The aim of this retrospective study is to assess, by examining medical records, whether repeated sedation with N_2_O for dental treatments improves the behavior of uncooperative children. The results seem to confirm that children’s cooperation improves significantly with repeated sedation regardless of age and the presence of intellectual or physical impairments.

The main strength of this study is the large sample of children considered. Few studies analyzed such a large sample [10,17] except for the study by Collado et al. [1], who evaluated a sample of 543 patients including both children and adults. Moreover, the sample considered in the present study includes both healthy children and those with intellectual or physical impairments, thus providing a comprehensive picture of the different types of patients who are routinely treated with N_2_O sedation in dentistry.

Inhaled nitrous oxide is considered safe and effective in reducing anxiety, providing analgesia and improving effective communication between child and dentist. The use of N_2_O at 50% concentration, such as that used on the present studied sample, is very common in pediatric dentistry [19]; however, the level of sedation achieved may not be sufficient to complete the dental treatment. Higher concentrations of up to 70% have been used [20]. Nevertheless, conflicting results on its safety have been reported [19,20]. A concentration of 60% was found to be more effective than 50% and safer than 70%. However, in the present study, because it was a retrospective study that drew on data obtained in children who were given a standardized protocol of 50% N_2_O, it is not possible to know whether a higher concentration of nitrous oxide would have achieved better results.

For the assessment of behavior, the modified Venham scale was preferred, as it is a sufficiently accurate tool for the behavioral staging of children based on objective reaction patterns [14]. The main instruments for measuring behavior include the Venham Scale [14], the Houpt Scale [21], the Ohio State University Behavioral Rating Scale (OSUBRS) [15] and the Frankl Scale [22]. These scoring systems are all widely accepted and validated. Although the OSUBRS scale may be considered the most accurate in describing patient behavior, it requires considerable skill and a learning curve for its use; in contrast, the Houpt and Venham scales are able to provide a simple and immediate assessment [15]. Another widely used tool to assess child behavior is the Frankl Scale [22]. The scale considers four behavioral categories, classifying child behavior as definitely positive, positive, negative and definitely negative. The scale is easy to use but allows a less detailed description of the child’s behavior than the Venham scale, which has six different scores instead of four.

The results of this study showed a decrease in Venham score during the administration of local anesthesia (T2) with the reiteration of sedation, as reported by previous studies, suggesting that this delicate phase can be facilitated with repeated sessions of dental care under conscious sedation [23,24]. Altered anxiety levels during the administration of local anesthesia have been reported [10,25]. This phenomenon may be traced to various causes, e.g., the sight of the needle, the shape of the syringe and the sensation of swelling and/or pressure on the soft tissue caused by the administration of the drug. To reduce these possible discomforts, the procedure could include a pre-application of a contact anesthetic and administration of the anesthetic with electronically guided devices [26]. Furthermore, in this study all treatments were performed by experienced pediatric dentists who are familiar with and implement behavioral techniques [4] that contribute to increasing the comfort of young patients even during difficult procedures such as local anesthesia. This aspect could be a limitation of the present study, as it is not possible to assess how much the behavioral methods contributed to the reduction of the Venham score. However, the anxiolytic effect induced by N_2_O seems to have made the memory of the dental treatment neither unpleasant nor frightening. In fact, patients at repeated sedation were more likely to cooperate readily with the procedure and to willingly have the nasal mask placed, as if seeking a pleasant sensation already experienced [16].

A statistically significant association between patient age and Venham score reduction was observed. Similarly, a recent study showed a success rate of 77.4% in primary school children and 96.7% among secondary school children [27]; furthermore, in the same paper, it was hypothesized that the success rate could be increased if patients were subjected to repeated sedation [27]. In contrast, a previous study including children aged between 3 and 6 years presented a lower Venham score compared to older children during a single N_2_O sedation [17]. This difference in the response to sedation between subjects of different ages could be due to the different number of sedations considered: repeated sedation could improve compliance in older children to a greater extent, as they are more able to learn from an experience they have already had.

The use of N_2_O sedation is recommended by the Italian guidelines for the management of patients with special needs requiring dental care [28]. In the present investigation, the efficacy of sedation with N_2_O in patients with systemic diseases that do not include intellectual impairment was comparable to that obtained in healthy patients. In contrast, in patients with an intellectual impairment, although there was an improvement in the Venham score, it was not comparable to that obtained by the rest of the sample. Patients with autism or psychiatric disorders showed less co-operation after induction of sedation than healthy patients, as reported in previous studies [23,29,30]. A possible cause of the reduced efficacy of N_2_O sedation in patients with an intellectual impairment might be due to the lack of nasal breathing, either due to the poor ability to breathe through the nose with the mouth open, or to unwanted behavior such as crying or talking. For these patients, the authors suggest a specific behavioral approach by trained clinicians in the management of patients with special needs.

The results of this retrospective study show that the repeated use of conscious sedation with N_2_O allows most uncooperative patients to reduce treatment-related anxiety, avoiding the use of general anesthesia. This positive effect of sedation on dental fear and psychological distress might remain five years after treatment, thus attributing an educational effect to N_2_O treatment [31].

A further possible limitation of this study is represented by the behavioral assessment tool used. Although the modified Venham scale is considered a reliable rating scale, it remains a subjective method that depends on the operator’s assessment. The salivary cortisol level has been suggested as an objective method of assessing patients’ discomfort; however, the effectiveness of this method requires further confirmation [32].

Furthermore, a number of variables that cannot be controlled in this type of study, such as the child’s individual characteristics, previous medical and dental experiences, the presence of toothache, in addition to parental anxiety, may have acted differently on each child, facilitating or hindering behavioral improvement [33].

The present results, even with the limitations of the data obtained from a retrospective study, may provide a solid basis for future prospective confirmatory studies. Further long-term studies would be interesting to help evaluate whether uncooperative patients undergoing repeated conscious sedations improve their collaboration enough to face dental treatments without sedation.

## 5. Conclusions

The results of the present retrospective study show that uncooperative children, with or without other medical conditions, can increase their trust in the dentist and their ability to cooperate with dental procedures thanks to conscious sedation with N_2_O, which can, especially if repeated, modify the anxious behavior of young patients.

## Figures and Tables

**Figure 1 ijerph-20-04037-f001:**
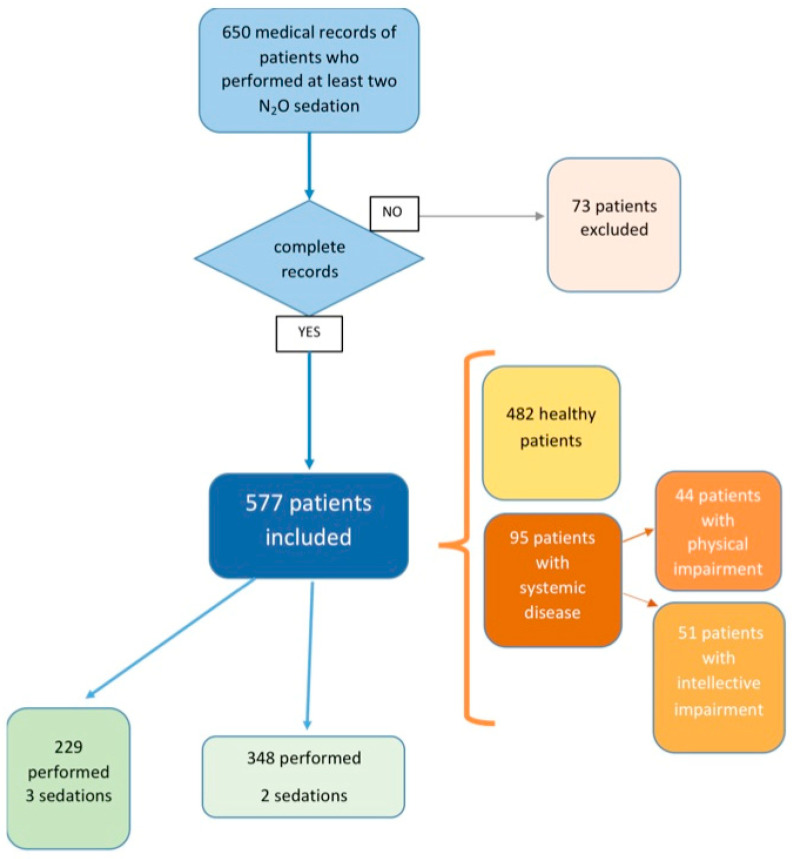
Flow chart of the study design.

**Figure 2 ijerph-20-04037-f002:**
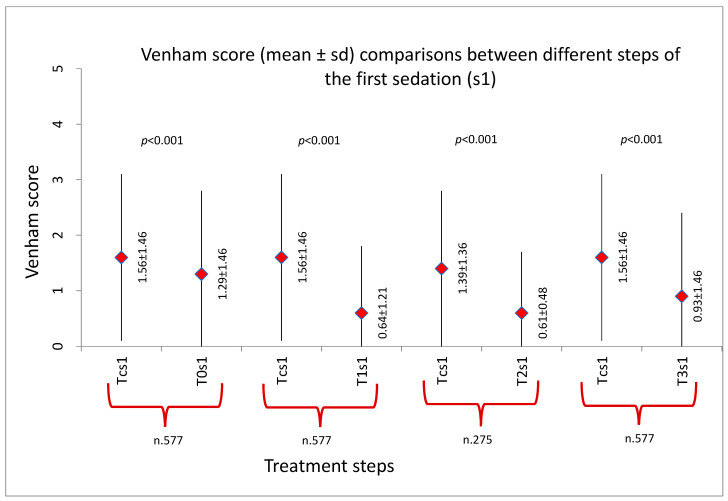
Venham score comparisons during the first session (s1). Tc: first contact with the dentist, T0: at the time of mask placement, T1: at least 3 min after the start of sedation, T2: during the administration of local anesthesia, T3: during the dental treatment. The sample size at step T2 for each sedation varies because not all patients received local anesthesia.

**Figure 3 ijerph-20-04037-f003:**
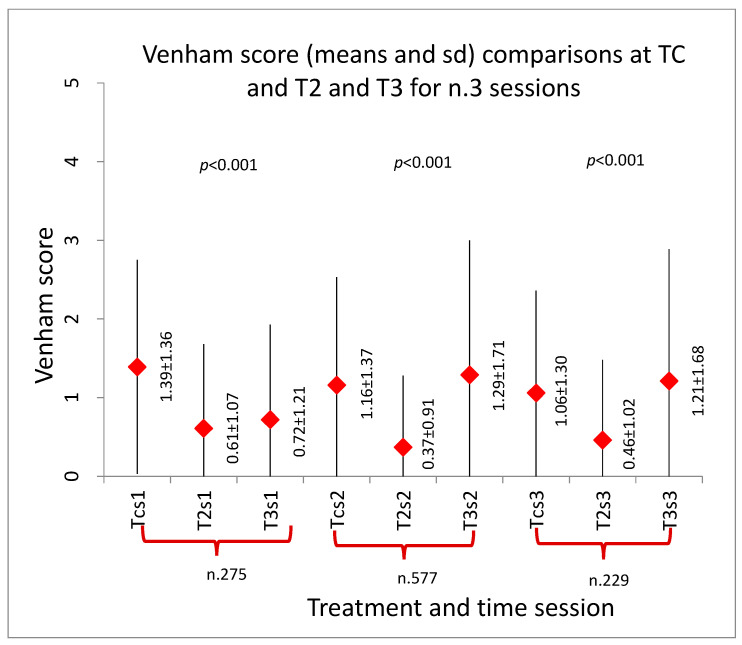
Venham score comparisons between Tc, T2 and T3 in the three sedations (s1, s2 and s3). Tc: first contact with the dentist. T2: during the administration of local anesthesia. T3: during the dental treatment.

**Figure 4 ijerph-20-04037-f004:**
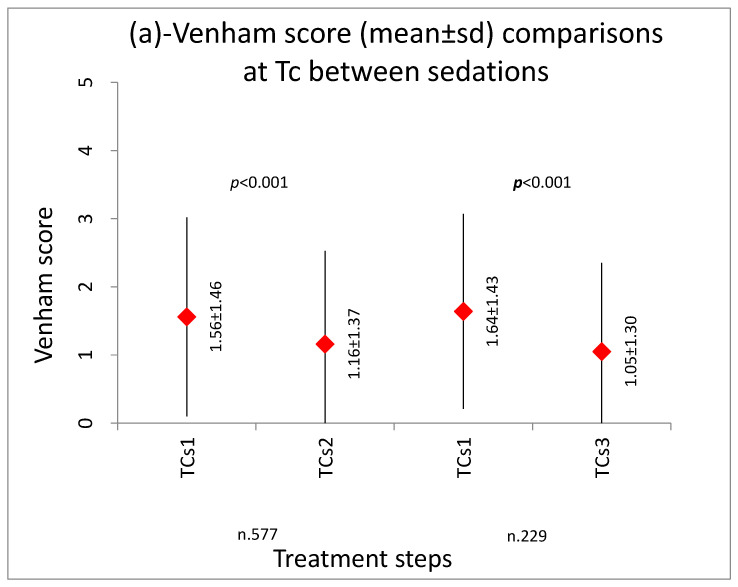
(**a**) Venham score comparisons at Tc (first contact with the dentist) and (**b**) Venham score comparisons at T0 (at the time of mask placement) between the first (s1) and the second sedation (s2) and between the first (s1) and the third sedations (s3).

**Table 1 ijerph-20-04037-t001:** Modified Venham scale.

0. Relaxed: smiling, willing, able to converse, displays behavior desired by the dentist.
1. Uneasy: concerned, may protest briefly to indicate discomfort, hands remain down or partially raised. Tense facial expression, high chest. Capable of cooperating.
2. Tense: tone of voice, question and answers reflect anxiety. During stressful procedure, verbal protest, crying, hands tensed and raised, but not interfering very much. Protest more distracting and troublesome. Child still complies with the request to cooperate.
3. Reluctant: pronounced verbal protest, crying. Using hands to stop procedure. Treatment proceeds with difficulty.
4. Interference: general crying, body movements sometimes needing physical restraint. Protest disrupts procedure.
5. Out of contact: hard loud crying, swearing, screaming. Unable to listen, trying to escape. Physical restraint require.

**Table 2 ijerph-20-04037-t002:** Characteristics of the included sample. (a) Demographic data and health conditions, (b) health conditions included intellectual impairment and (c) diseases not involving the intellectual disability.

(a)	
**Variables**	**n (%)**
**Age**	
3–6 years	367 (63.6)
7–14 years	210 (36.4)
**Sex**	
Female	268 (46.5)
Male	309 (53.6)
**Health conditions**	
Healthy	482 (83.5)
Physical impairment	44 (7.6)
Intellectual impairment	51 (8.8)
(b)	
Down’s syndrome	22
Autism Spectrum Disorders	13
Attention Deficit Hyperactivity Disorder (ADHD)	7
Cognitive disability	6
Rare genetic syndromes	3
**Total number**	**51**
(c)	
Cardiac disease	18
Growth impairment	12
Epilepsy	6
Leukemia	4
Other health conditions	4
**Total number**	44

## Data Availability

The datasets analyzed during the current study are not publicly available but are available from the corresponding author on reasonable request.

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
