# Peer review of "Behavioral Modifications in Children after Repeated Sedation with Nitrous Oxide for Dental Treatment: A Retrospective Study"

_ijerph, 2023, doi:10.3390/ijerph20054037_

Round 1
Reviewer 1 Report
The authors reported a retrospective study with the aim to evaluate the influence of repeated sedation with nitrous oxide for dental treatment on behavior modification in children. The topic is up to date but there are a few remarks.
Major remarks:
Some factors that may influence cooperative behaviors in children such as anxiety, parenting style and treatment duration were not included in the study and not assessed.
Page 2/lines 83-85: “In the sedation session, heart rate, oxygen saturation, and blood pressure are monitored regularly at the beginning of the dental treatment, every 10 minutes during the procedure, and at the end of the treatment.“
Comment: If all these parameters were monitored, they should be presented and interpreted in the Results and Discussion sections.
Table 2. Comment: Abbreviation ADHD should be explained
Figure 3. Comment: It would be interesting to include T3 scores in the Figure and compare them to Tc and T2 scores. Scores during dental treatment are not less important than the scores during the administration of local anesthesia.
Figures 4a and 4b. Comment: These two figures are absolutely the same. The score values are the same in both Figures. Figure 4b does not present Venham score comparisons at T0.
Page 8/lines 191-205. Comment: These paragraphs should be completely revised and results of statistical tests and analyses should be presented.
Discussion section
Comment: Discussion section requires revision. Some new references, that are published in last two to three years, are missing. In my opinion discussion regarding effectiveness of different dosages of nitrous oxide, possible complications and adverse reactions of nitrous oxide/oxygen inhalation sedation and the use of modifications of Frankl Behavior Rating Scale as an instrument for behavior evaluation in pediatric dentistry should be included in the manuscript.
Author Response
Some factors that may influence cooperative behaviors in children such as anxiety, parenting style and treatment duration were not included in the study and not assessed.
The reviewer is correct in stating that several factors related to the child's character, experiences, and not least the empathy of the practitioner, as well as aspects related to the family may play a role in the child's ability to cooperate with therapy. However, as reported in the paper, all children undergoing sedation were untreatable, in the opinion of an experienced pediatric dentist, without resorting to this method of behavior management. Since, as far as the authors know, no strategy other than sedation was implemented to modify behavior regardless of the reasons for noncooperation, it is possible to assert that an important role was played by sedation.
Taking up the reviewer's suggestion, we have added the following sentence in the discussion section: “Furthermore, a number of variables that cannot be controlled in this type of study, such as the child's individual characteristics, previous medical and dental experiences, the presence of toothache in addition parental anxiety, may have acted differently on each child, facilitating or hindering behavioral improvement”
Page 2/lines 83-85: “In the sedation session, heart rate, oxygen saturation, and blood pressure are monitored regularly at the beginning of the dental treatment, every 10 minutes during the procedure, and at the end of the treatment. “Comment: If all these parameters were monitored, they should be presented and interpreted in the Results and Discussion sections.
While understanding the reviewer's request, the choice to consider only the behavioral assessment parameter was dictated by the following factors:
the present is a retrospective study and, although the parameters mentioned were available in the patients' medical records, according to the authors they did not have to be considered in the present study because they had little influence on the outcome under investigation, i.e. the cooperation offered. Indeed, the variables measured during the procedures, such as heart rate and blood pressure, are mainly related to a pharmacological effect of nitrous oxide, but only partly explain the patient's behaviour. In fact, conscious sedation has an action on these parameters but is unable to change the patient's will if he or she opposes the therapy, as shown by the reduced results obtained on patients with intellectual disabilities. These limitation of N2O sedation were recently reported in a metanalysis (J Dent Anesth Pain Med. 2021;21:527-545) Indeed, one of the essential aspects for successful inhalation sedation is nasal breathing, which is difficult to achieve if the patient is crying and agitated. The latter category of patients may need longer training to learn to cooperate than the two or three sessions of sedation considered in this analysis.
Table 2. Comment: Abbreviation ADHD should be explained Thanks for the correction, we have made the change
Figure 3. Comment: It would be interesting to include T3 scores in the Figure and compare them to Tc and T2 scores. Scores during dental treatment are not less important than the scores during the administration of local anesthesia. Thanks for the suggestion, we proceeded to insert a figure 3 supplemented with the requested data
Figures 4a and 4b. Comment: These two figures are absolutely the same. The score values are the same in both Figures. Figure 4b does not present Venham score comparisons at T0.
We are sorry for the mistake, we proceeded to insert a figure 4b corrected with the requested data
Page 8/lines 191-205. Comment: These paragraphs should be completely revised and results of statistical tests and analyses should be presented. Thanks for the suggestion, we have integrated this section with the results of the statistical analysis
Discussion section
Comment: Discussion section requires revision. Some new references, that are published in last two to three years, are missing. In my opinion discussion regarding effectiveness of different dosages of nitrous oxide, possible complications and adverse reactions of nitrous oxide/oxygen inhalation sedation and the use of modifications of Frankl Behavior Rating Scale as an instrument for behavior evaluation in pediatric dentistry should be included in the manuscript
We thank the reviewer for suggestions.
Some new references have been added, and, albeit briefly, both the different protoxide concentrations used and the Frankl scale as a useful tool for behavior assessment have been described.
We did not go into adverse effects because this is not a parameter that was collected by the clinical records. However, we would like to point out that no adverse effects, except for a few episodes of nausea, were reported by patients.
Reviewer 2 Report
Dear Authors
I have read your manuscript with great interest. I agree with you that there is very little information about this item. Having had different groups of children and considering those with physical impairment and intellectual impairment was very appropriate. This information will help to convince many other institutions in other countries that refuse to use sedation in children.
Please only make a few minor adjustments to the manuscript.
1.- Unify how you are going to refer to "nitrous oxide" with words or with its formula. In principle, line 36 is where you should write it between parentheses and not until line 38. From there, you should write "N2O" throughout the document since you write it both ways, for example:
"Nitrous oxide" in lines: 52, 54, 64, 73, 81, 86, 207, 218, 269, 272,
"N2O" in lines: 58, 106, 209, 257, 263, 286
2.- Add some references to some of the statements you make. For example
Lines 42, 47, 89,
3.- Add in parentheses the registration number assigned by ClinicalTrials.gov
4.- Explain what "ADHD" means in Table 2
5.- In line 228 what you want to say is "Vanhee et. al"?
Author Response
1.- Unify how you are going to refer to "nitrous oxide" with words or with its formula. In principle, line 36 is where you should write it between parentheses and not until line 38. From there, you should write "N2O" throughout the document since you write it both ways, for example:"Nitrous oxide" in lines: 52, 54, 64, 73, 81, 86, 207, 218, 269, 272, "N2O" in lines: 58, 106, 209, 257, 263, 286 Thanks for the correction, we have made the changes
2.- Add some references to some of the statements you make. For example
Lines 42, 47, 89,
References were added as suggested
3.- Add in parentheses the registration number assigned by ClinicalTrials.gov
The protocol for the present study has been submitted to ClinicalTrials.gov, but a registration number has not yet been issued
4.- Explain what "ADHD" means in Table 2 Thanks for the correction, we have made the change
5.- In line 228 what you want to say is "Vanhee et. al"? Thanks for the correction, we have made the change
Reviewer 3 Report
Dear authors!
The chosen study design, aimed at assessing the effect of repeated sedation with nitrous oxide on the degree of children's collaboration, corresponds to the objective.
Statistical analysis of the data in the submitted manuscript was carried out correctly, using preliminary data verification. However, minor clarifications are required. Specific comments are given in the attached file.

Author Response
Lines 132-134 Please specify the method of calculating the sample size. Was the calculation made in the program or manually? Thanks for the suggestion, we have integrated this section, the sample size calculation was planned on a measure of the mean reduction effect of the Venham score. The calculation was carried out with the software Stata 17.
Lines 144 In the "Materials and Methods" section, you indicated that a sufficient number of patients for the study corresponds to 100. Why did not you stop at this value and exceed the sample size by 5 times?
It might also worth considering the effect size to calculate the sample size. This parameter is also important for your work citing. Systematic review and meta-analysis often take into account the effect size in the publication selection criteria. Thanks for the remark, the sample size consisting of 100 subjects guaranteed to estimate a reduction effect of 1 mean point of the Venham score. To guarantee this number between two sessions we had to increase the size of the starting sample because the number of subjects is reduced in the sessions following the first
Line 148 The clarification "(standard deviation)" is unnecessary, since in the "Materials and Methods" section you have already described a way of presenting descriptive statistics. Thanks for the correction, we have made the change
Table 2. In the table header the number of patients is indicated by an uppercase letter ("N"). It is necessary to correct to lowercase ("n"). Thanks for the correction, we have made the change
Lines 270-272 Please specify how exactly the data of the literary reference No 24 reinforce this fragment of the text?
The mentioned study concludes that literature data are insufficient and, consequently, the conclusion is not unambiguous.
We apologise for the mistake, the reference was not cited correctly. It was modified
Round 2
Reviewer 1 Report
The manuscript has been modified according to the suggestions and improved according to my point of view.